# Inter-Day Reliability and Changes of Surface Electromyography on Two Postural Muscles Throughout 12 Weeks of Hippotherapy on Patients with Cerebral Palsy: A Pilot Study

**DOI:** 10.3390/brainsci10050281

**Published:** 2020-05-06

**Authors:** Hélène Viruega, Inès Gaillard, Laura Briatte, Manuel Gaviria

**Affiliations:** Institut Equiphoria, Combo Besso-Rouges Parets, 48500 La Canourgue, France; helene.viruega@equiphoria.com (H.V.); ines.gaillard@equiphoria.com (I.G.); laura.briatte@equiphoria.com (L.B.)

**Keywords:** cerebral palsy, postural balance, postural muscles, EMG, neurorehabilitation, hippotherapy

## Abstract

Cerebral palsy (CP) is an umbrella term covering a group of permanent developmental disorders of movement and posture characterized by highly variable clinical features. The aim of this study was to assess the short-term and mid-term effects of neurorehabilitation via hippotherapy on the contractile properties of two key postural muscles during functional sitting in such patients. Thirty-minute hippotherapy sessions were conducted biweekly for 12 weeks in 18 patients (18.1 ± 5.7 years old). Surface electromyography (EMG) was implemented bilaterally in rectus abdominis and adductor magnus. We quantitatively analyzed the amplitude of EMG signals in the time domain and its spectral characteristics in the frequency domain. EMGs were recorded at the beginning and end of each session on day one and at week six and week twelve. Statistical analysis revealed a substantial inter-day reliability of the EMG signals for both muscles, validating the methodological approach. To a lesser extent, while beyond the scope of the current study, quantitative changes suggested a more selective recruitment/contractile properties’ shift of the examined muscles. Exploring postural control during functional activities would contribute to understanding the relationship between structural impairment, activity performance and patient capabilities, allowing the design of neurorehabilitation programs aimed at improving postural and functional skills according to each individual’s needs. The present study provides basic quantitative data supporting the body of scientific evidence making hippotherapy an approach of choice for CP neurorehabilitation.

## 1. Introduction

Cerebral palsy (CP) is a motor syndrome that results from a permanent, non-progressive injury in the developing brain. The clinical picture is shaped according to the time of occurrence of the lesion, the location and severity of the neural alterations, life experience, and activity level of the subject. CP includes a broad number of movement and postural disorders. The severity, the type of motor impairment, and the associated deficiencies (e.g., communication, intellectual capacity, behavioral problems) are highly variable. Overall prevalence has remained stable over the past 40 years, despite progress in antenatal and perinatal care. In France, around 125,000 people are affected [1].

Postural control is a prerequisite for activities of daily living (ADL). Generally, upholding a stable postural balance against gravity is challenging and requires a refined network organization, i.e., accurate connections between sensory inputs, central CNS integration, and muscle effector outputs. Dynamic postural control includes feedforward or anticipatory postural adjustments that predict disturbances and produce preprogrammed responses to maintain stability, and feedback or compensatory postural adjustments activated by sensory events following loss of postural balance. Both mechanisms are centrally driven to optimize postural balance [2,3]. Fine-tuning of postural muscle contraction during specific motor tasks is inefficient in most individuals with CP, since flawed co-activation of antagonist muscles as well as difficulties in temporal organization and amplitude modulation of the muscles are present [4]. In patients with CP, one of the main clinical dysfunctions consists of the failure to activate the postural muscles in the right order and to fine-tune them [5]. Other neurological issues limit the control of balance in patients with CP, such as spasticity, hyperactive stretch reflexes, and dyskinesia [6].

Neurorehabilitation of postural balance in patients with CP is rather complex, as the clinical shapes among these patients are extremely heterogeneous. Generally speaking, unstable support surfaces induce increased activation of the trunk musculature and a constant muscle response to adjust the posture against instability. As a result, patients do not need to carry out complex exercise routines to benefit from trunk rehabilitation on these surfaces [7,8]. Hippotherapy can be considered an unstable support therapy that generates up to 100 three-dimensional smooth physical trunk and girdle micromovements per minute, mimicking the patient’s body during walking [9]. Moreover, the therapy is released in a particularly appealing and friendly ecosystem where children are generally confident and highly motivated. This general panorama plays a key role in optimizing brain plasticity through a key enriched environment for the promotion of functional recovery of postural balance [10].

From the effector point of view, when balance of a seated person is challenged, postural stability is restored via activation of the major trunk muscles. Even during quiet sitting, tonic muscle activity can be observed in all major trunk muscles, but tonic activity is not sufficient to resist perturbations. Additional phasic muscle activity, which is symmetrical across the body, is necessary to slow the movement of the trunk and bring it back to a vertical position. However, each trunk muscle responds to multiple, but not all, perturbation directions. In spite of the crucial role of those mechanisms for postural balance, the muscular response to seated perturbations and its characteristics are not fully understood or described in health or disease states [11,12].

Assessing postural control during functional tasks is important in order to understand balance adjustments adopted by CP individuals in their daily lives. Since hippotherapy might support the regularization of sitting postural balance [10], we aimed to explore the role of the pelvic girdle in this regularization by choosing two key muscles above and below this main anatomical structure, namely, the rectus abdominis (RA) and the adductor magnus (AM). RA participates in sitting stability by controlling the backward–forward sway of the trunk [13]. AM is both a dynamic stabilizer of the pelvis and femur as well as a prime mover of the femur into adduction and has the potential to affect postural control of the pelvis while sitting [14,15]. Since one of the main neurophysiological alterations of the CP neuromotor system is the activation and fine-tuning of the postural muscles, we focused on the recording of the electrical activity of these muscles by means of surface electromyography (EMG). We deliberately chose to analyze these two non-antagonistic postural muscle groups, nevertheless with opposite actions in pelvic positioning (retroversion and anteversion, respectively). Indeed, the complex multilayered arrangement of extensor muscles of the lumbar region (multifidus, longissimus thoracis, iliocostalis lumborum) makes the interpretation of muscle activation through surface EMG during postural tasks difficult [16]. On the other hand, even if the individual hip adductor muscles are quite close together at the medial side of the proximal thigh (in particular adductor magnus, gracilis, and adductor longus), a good reliability of surface EMG can be demonstrated when recording the activity of the AM muscle [17]. In addition, AM plays a role in hip extension, contributing to the stabilization of the pelvic girdle during riding. In fact, the hip extension moment arm length of AM changes with the hip angle and is a more effective hip extensor than either the hamstrings or gluteus maximus when the hip is flexed.

In this pilot, our objective was to strengthen our previous hypothesis regarding the reinforcement of postural balance [10] by drawing a general picture of the changes in the contractile properties of key postural muscles through hippotherapy. We characterized the initial EMG patterns in the time and frequency domains and their modification during a 12-week hippotherapy program for neurorehabilitation with the aim of paving the way for future clinical trials in the neurorehabilitation field. 

## 2. Materials and Methods

### 2.1. Subjects and Inclusion Criteria

Eligible patients included males and non-pregnant females from 5 to 25 years old, presenting a sensorimotor impairment secondary to CP, and able to understand the basis of the experimental protocol. The patients were recruited among the residents of the Montrodat Center for Rehabilitation Medicine (48,100 Montrodat, France).

The study was carried out in agreement with the recommendations of the regional ethics committee of Montpellier-France (CPP Sud-Mediterranée IV; n°150403 dated 06/09/2015), who provided ethical approval, and institutional and national guidelines for research in human individuals (ANSM approval n°151108B31 dated 09/02/2015). Patients were free to withdraw at any time without impacting the quality of their care. All subjects and/or their parents/legal representatives/home institutions gave written informed consent in accordance with the Declaration of Helsinki.

The inclusion criteria were as follows:Diagnosis: CP consisting of spastic tetraparesis, diparesis, monoparesis, or hemiparesis;Degree of impairment according to GMFCS: Levels I to IV (walking without limitation to self-mobility with limitations/may use powered mobility);Good comprehension ability (no cognitive or behavioral impairment impeding the patient to follow a normal school curriculum);A minimal abduction of the hip of 25 degrees bilaterally with no history of hip dislocation and/or dysplasia;No concomitant pathology that may impair sensorimotor and/or cognitive function;Certificate of non-contraindication issued by the responsible physician.

The exclusion criteria were as follows:Patient or parent/legal representative who did not give written informed consent;History of epilepsy uncontrolled by medication;Allergic reaction to horses, dust, or self-adhesive electrodes (surface EMG);Recent surgical history (<1 year) or surgery corresponding to a specific procedure incompatible with riding, e.g., selective dorsal rhizotomy, extended spinal arthrodesis;History of horseback riding or hippotherapy treatment during the last six months before recruitment;History of treatment with botulinum toxin over the six months preceding the start of the protocol;Therapeutic intervention planned during the duration of the study (injection of botulinum toxin, tenotomy, tendon transfer, etc.);History of uncontrolled pain;Pregnant or lactating women;Patients participating in another biomedical research or in a period of exclusion.

Eighteen subjects (7 females and 11 males, aged 18.1 ± 5.7 years) with CP diagnoses were included in this study. Subjects were recruited based on inclusion criteria, all with moderate to severe alterations in muscle tone as well as impaired postural balance, thus hindering their ADL. Standing and walking were impaired at varying degrees (Table 1).

### 2.2. Study Design

We used a single-group prospective ancillary longitudinal pilot study to evaluate the effects of an experimental protocol of hippotherapy in the activation and modulation of the amplitude of muscle responses during dynamic postural balance in patients with CP. The study took place at the Equiphoria Institute (La Canourgue, France: https://www.equiphoria.com/en/). As a reminder, hippotherapy is a rehabilitation method achieved through the movement of a horse at a walk. It is a dynamic activity where the amplitude of movement of the patient’s body is similar to the human walking activity (micro-movements of postural muscles). The hippotherapy protocol consisted of two sessions a week, separated by at least two days rest interval, for a period of 12 weeks, i.e., 24 hippotherapy sessions, according to the parent study. The protocol shape was decided based on our own clinical experience, as well as taking into consideration that evidence-based hippotherapy practice elsewhere has not been standardized and is quite heterogeneous, i.e., from 6 to 12 weeks in weekly, biweekly, or triweekly bases [18,19,20,21].

The physical therapy exercises performed during sitting on the horse at a walk consisted of three phases: (i) 5 min of warm-up by passive and active mobilization of the lower limbs, and passive and active stretching of the different muscle groups; (ii) 20 min of work on sitting posture, balance and fine-tuning reactions (eyes open and closed), work of fine motor skills of the upper limbs by manipulating objects, strengthening of different muscle groups, and practiced activities to reinforce internal body image [22]; and (iii) 5 min of relaxation with passive mobilization and passive stretching, especially of the flexor muscles. Standing and walking was not needed for mounting the horse during hippotherapy; the institute was equipped with a ramp, allowing a wheelchair to be placed alongside the horse. Our trained staff operated the transfer of non-walking patients. Once on the horse, the patient was able to maintain an adequate postural balance. The therapist walking alongside the horse during the session was an occasional support if needed. The horse was equipped with a pad (no saddle or stirrups) and a vaulting surcingle, allowing maximum contact of the patient with the horse’s back and freedom of movement. While stirrups are useful in certain circumstances for stabilizing some patients, they were not used in our study as stirrup support can provoke pelvic retroversion and therefore can restrict pelvic movement with consequently limited postural adjustment possibilities. 

Sample size calculation (GLIMMPSE 3.0 open access software) was based upon 80% power for a comparison of one-group repeated measures with a type I error rate of 0.05 using the Hotelling–Lawley Trace multivariate approach, supposing a mean scale factor of 1 (unchanged mean values over time) and a variability scale factor of 0.5 (reduction by half of the SDs over time). The total sample size was n = 15.

### 2.3. Protocol

Three measurement times were determined throughout the neurorehabilitation protocol: (i) the baseline at the first session, (ii) an intermediate value corresponding to the sixth week of treatment, and (iii) an end value at the last session at the twelfth week of treatment. Two recordings were made each time, one at the beginning of the session (first 3 min) and one at the end of the session (last 3 min) while the patient was on the horse. The patient’s position on the horse pad did not cause any pressure or unsticking of the self-adhesive electrodes of the AM muscles. Two patients performing their session simultaneously were recorded at the same time. We measured the modulation of the amplitude of muscle response from an electrophysiological point of view. To this end, self-adhesive surface electrodes were placed symmetrically and bilaterally on the RA muscles (right and left) and AM muscles (right and left), in accordance with placement standards [19]. The guidelines for electrode placement outlined by SENIAM (2009) [23] could not be followed in our case, since RA and AM are not included in the specific electrode placement recommendations. We therefore used placement usually recommended in comparable studies among the reviewed literature. For the RA muscle, the electrodes were placed 3 cm lateral to the umbilicus with the lower border of the caudal electrode at the level of the umbilicus, both oriented parallel to the muscle fibers [24,25]. For the AM muscle, a point was selected in the proximal third along a line from the tuberculum pubis to the epicondylus medialis femoris with the lower border of the caudal electrode at this point level and both electrodes also oriented parallel to the muscle fibers [26,27]. Due to the symmetrical placement of the surface electrodes, taking into account precise anatomical landmarks in each individual, we aimed to compare right/left asymmetry.

Except for placement, we followed the SENIAM recommendations for the EMG sensor’s general placement procedure [23]. Prior to electrode placement, the skin was shaved when necessary and gently cleaned with 70% alcohol. A wireless hardware system (FREEEMG 300, BTS Bioengineering, Milan, Italy) was used for dynamic surface EMG, with four analog inputs per patient (eight total inputs, i.e., two patients simultaneously), an A/D converter with 16-bit resolution, and a common rejection mode ratio of >100 dB at 50–60 Hz and 20–500 Hz bandpass filter. The sampling frequency was 1000 Hz per channel. Pre-gelled disposable self-adhesive differential bipolar Ag/AgCl surface electrodes measuring 1 cm in diameter each (Kendall, Covidien, Germany), spaced 2 cm from center to center, were connected to the portable amplifier, with an output impedance of >10 MΩ, a sensitivity of 1 μV, and a gain of × 1000. The transmission frequency was done through a 2.4 GHz ISM band (standard IEEE 802.15.4). Subjects were sat on the massage table with the trunk straightened axially and the legs slightly extended and wide apart to obtain around 130 degrees of hip flexion and 45 degrees of abduction. All electrodes were placed by the same experimenter throughout the study. A test was done to determine whether the electrodes were placed properly on the muscle and connected to the wireless WiFi standard 802.11b receiving unit workstation before starting the session. Wireless real time data transmission allowed for an acquisition range of up to 20 m in an open space without obstacles (in this case, a riding arena of 100 m^2^). The signal processing was carried out using dedicated software, the BTS EMG-Analyzer, including predefined templates for clinical and research assessments and a graphical interface. The sampling frequency was 1 kHz and the resolution was 16 bits, allowing a high-quality acquired signal, low noise, and absence of movement artifacts. Data were stored on a dedicated laptop for delayed treatment and analysis.

### 2.4. EMG Artifact Considerations

Although surface EMG was used to detect changes in activity of postural muscles during long-term recordings, there were a number of problems associated with EMG recordings of postural muscles, such as cross-talk, noise, and a relatively low signal-to-noise ratio (SNR). The level of muscle activation during postural activity can be very low, sometimes less than 5% of maximal voluntary contraction. When muscle activation is at such a low level, an acceptable SNR can only be obtained by markedly reducing the noise level in various ways [28]. On the one hand, improvement in the design of modern amplifiers and EMG integrated systems has reduced noise magnitude, which is the case of the FREEEMG 300. On the other hand, a potentially large source of noise is that caused by the electrode–skin interface. While thermal noise is frequently assumed to be the major contributor to noise from the electrode–skin interface, studies showed that thermal noise can account for as little as 10% of the electrode–skin noise signal. Moreover, proper skin preparation obtained by shaving the electrode site and thorough cleaning with alcohol (SENIAM recommendations) results in low skin impedance over time and allows a stable EMG signal during long-term postural recordings [28].

### 2.5. Outcome Measures and Data Analysis

The nature of the EMG signal makes the statistical analysis of this signal the most appropriate method in the quantitative study of muscle electrical activity during voluntary contraction [29,30]. Relying on previous reference studies, the analysis of the signal was carried out essentially in two domains, time and frequency. The statistical processing of the signal in these two domains made it possible to define a number of reliable parameters, informing us on the variations of muscle electrical activity during activation [31,32,33,34,35,36,37,38].

Time-domain analysis of EMG amplitudes is frequently used as a muscle force detection tool. Most common EMG amplitude variables are the rectified EMG (iEMG), the peak root mean square (pRMS), and the mean root mean square (mRMS), which mainly reflect the number of active motor units of the muscle and their discharge rates during voluntary contraction. Among them, the mRMS is the most robust measure of EMG amplitude with respect to motion artifacts, signal noise, and temporal changes in movement. The RMS amplitude is more representative of muscular activity because it does not depend on the direction of the muscle fibers with respect to the electrodes, therefore, this is one of the most used time variables [33,37,39]. 

On the other hand, frequency-domain analysis of the EMG power spectrum was employed to detect neuromuscular impairment. Changes in spectral characteristics are related to the synchronization of several motor units or changes in the rate of recruitment and discharge of the motor units, such as during muscle fatigue [36]. Two power spectrum parameters provide useful measurements of the EMG spectrum and are the most reliable, namely, the median frequency (MDF) and the mean frequency (MNF). Among these, MDF was found to be the least sensitive to noise, which is particularly useful when the signal is obtained during low-level contractions and from signal-to-noise [33].

We retained one parameter in the time domain (mRMS) and two in the frequency domain (MDF and MNF) to characterize the muscular function of the chosen muscles. We decided to focus on the raw EMG, since normalization through the use of the maximal voluntary isometric contraction was not possible for all our patients. Indeed, some experienced difficulty in eliciting and modulating voluntary contraction of at least one of the studied muscles. Moreover the choice of our methodology was reinforced by the fact that our aim was to analyze the EMG signal frequency content and the activation and amplitude of a given muscle between short-term interventions of an individual under the same experimental conditions, i.e., without changes to the EMG set-up [40,41].

A preliminary step and prerequisite for the longitudinal characterization of the myoelectric signals was the confirmation of the reliability of the repeated measures through the protocol timeline. We therefore calculated the intraclass correlation coefficient (ICC) of the absolute value of RMS. Shrout and Fleiss suggested that the two-way mixed-effects model was appropriate for testing intra-rater test–retest reliability with multiple scores from the same rater. The ICC_3,k_ was therefore calculated according to the method of Shrout–Fleiss [42]: $${\mathrm{ICC}}_{3,k}=\frac{{\mathrm{MS}}_{S}-{\mathrm{MS}}_{E}}{{\mathrm{MS}}_{S}}$$
where ${\mathrm{MS}}_{S}$ means subject’s mean square and ${\mathrm{MS}}_{E}\;$means error mean square.

In summary, we analyzed inter-day reliability, symmetry, short-term effects (immediate impact of the session), and long-term effects (cumulative mid-term impact) of hippotherapy on the contractile properties of two postural muscles using raw surface EMG signals. 

### 2.6. Statistical Analysis

Values were presented as means and standard deviations and corresponding values at the lower and upper limits of the 95% confidence interval. One-way analysis of variance (ANOVA), each line representing a repeated measure, was used to (i) compute the intraclass correlation coefficient (ICC_3,k_) to corroborate the inter-day reliability of the measures and (ii) compare the data followed by a Tukey multiple-comparison post-test to determine the presence of left–right asymmetry, the impact of the session (immediate short-term effect), and the impact of the overall treatment (cumulative mid-term effect) on the retained values (mRMS, MNF, and MDF). We applied a Geisser–Greenhouse correction, assuming that the variability of the differences was not equivalent (sphericity). A *p*-value below 0.05 was considered statistically significant. GraphPad Prism 8 software (GraphPad Software Incorporated, San Diego, CA, USA) was used for statistical analysis.

## 3. Results

### 3.1. Inter-Day Reliability of the Absolute Amplitude of Surface EMG

The reliability of absolute RMS across the sessions was computed (day 1 versus week 6 versus week 12) using the intraclass correlation coefficient (ICC_3,k_). The ICC_3,k_ values were interpreted using the categories proposed previously, in which an ICC between 0.00 and 0.20 was considered poor, 0.21 and 0.40 was fair, 0.41 and 0.60 was moderate, 0.61 and 0.80 was substantial, and 0.81 and 1.00 was almost perfect [43].

We calculated the ICC_3,k_ in nine individual measurements at the beginning of the session for RA and AM and obtained the following values.

For RA muscle, intraclass correlation coefficient (ICC_3,k_) = 0.68 based on the following statistics (Table 2):

For AM muscle, intraclass correlation coefficient (ICC_3,k_) = 0.70 based on the following statistics (Table 3):

We concluded that there was substantial reliability of the surface EMG measurements of the muscles.

### 3.2. Temporal Analysis

#### 3.2.1. Symmetry

The amplitude of the EMG signal, which was measured using the mRMS, did not show significant differences related to laterality (ANOVA followed by the Tukey multiple-comparison test: *p* = 0.1692, F(2.31, 28.3) = 1.86, DF = 11 for RA and *p* = 0.2352, F(2.40, 27.5) = 1.52, DF = 11 for AM). Muscular contraction was substantially symmetrical regardless of the muscle (RA, AM), the time of the recording (beginning and end of the session), and the time of treatment (T0, week 6, week 12) (see Table 4).

Since no statistical differences nor a specific trend were noticed between the right and left sides, we decided to pool the respective mRMS values for the subsequent analysis, i.e., effect of the session (3.2.2) and effect of the overall treatment (3.2.3).

#### 3.2.2. Effect of the Session (Short-Term Effect)

The effect of the session on the amplitude of the EMG signal, measured using the mRMS, was not significantly different for the RA muscle (ANOVA *p* = 0.0622, F(1.43, 35.1) = 3.33, DF = 5; Tukey multiple-comparison test not significant for the relevant values), but was found to be significant for the AM muscle (ANOVA *p* = 0.0303, F(2.56, 57.4) = 3.38, DF = 5; Tukey multiple-comparison test not significant for the relevant values). The amplitude of the EMG signal (reflecting the activation of the muscle through mRMS) was substantially equivalent in the RA muscle during each session but showed a clear decreasing trend in the AM muscle (see Table 5 and Figure 1).

#### 3.2.3. Effect of the Overall Treatment (Mid-Term Effect)

The effect of the overall treatment on the amplitude of the EMG signal, measured using the mRMS, was not significant for the RA muscle (ANOVA *p* = 0.0622, F(1.43, 35.1) = 3.33, DF = 5; Table 4). However, when comparing the mRMS values in pairs using the Tukey multiple-comparison test, a significant difference (*p* = 0.036) of the mRMS value during the first three minutes of the session was noticed between the first and last day of hippotherapy treatment (see Figure 2). Also, when comparing the mRMS value of the first three minutes of the first session with the mRMS value of the last three minutes of the last session, a significant difference (*p* = 0.024) was observed.

On the other hand, mRMS values were significantly different for the AM muscle (ANOVA *p* = 0.0303, F(2.56, 57.4) = 3.38, DF = 5; Table 4). When comparing the mRMS values in pairs using the Tukey multiple-comparison test, a significant difference in the mRMS values was shown between the first three minutes of the initial (T0) and final (week 12) sessions (*p* = 0.036), and between the first three minutes of the initial (T0) and intermediate (week 6) sessions (*p* = 0.018) (see Figure 3). Also, when comparing the mRMS value of the first three minutes of the first session with the mRMS value of the last three minutes of the last session, a significant difference (*p* = 0.004) was also evident.

Overall, the amplitude of the EMG signal (reflecting the activation of the muscle) showed a decreasing and more homogeneous trend (as evidenced by the decrease in standard deviation and confidence intervals) during the treatment, regardless of the muscle group (see Table 6 and Figure 1, Figure 2 and Figure 3).

### 3.3. Frequency Analysis

#### 3.3.1. Symmetry

The frequency spectrum of the EMG signal, measured through the MNF and the MDF, did not show significant differences when comparing the right and left corresponding muscles (ANOVA *p* = 0.2246, F(3.37, 41.9) = 1.50, DF = 11 for MNF and *p* = 0.2437, F(3.60, 44.8) = 1.43, DF = 11 for MDF for the RA muscles; *p* = 0.4667, F(3.49, 43.2) = 0.890, DF = 11 for MNF and *p* = 0.3238, F(3.79, 47.3) = 1.20, DF = 11 for MDF for the AM muscles). The frequency spectrum of the muscular contraction was substantially symmetrical regardless of the muscle, the time of recording (first three minutes and last three minutes of the session), and the time of the treatment (T0, week 6, week 12) (see Table 7 and Table 8). The mean RA asymmetry reached up to 11.2% for MNF and 16.3% for MDF, and was within the confidence interval of the group values; the mean AM asymmetry reached up to 1.1% for MNF and 1.8% for MDF, and was also within the confidence interval of the group values.

The symmetry in the response of both sides for each muscle, RA and AM, allowed us to pool the respective MNF values (right and left) and MDF values (right and left) for the subsequent analysis, i.e., effect of the session (3.3.2) and effect of the overall treatment (3.3.3).

#### 3.3.2. Effect of the Session (Short Term Effect)

The effect of the session on the frequency spectrum of the EMG signal, measured through MNF and MDF, was not as a whole significant regardless of the muscle (ANOVA *p* = 0.1709, F(2.69, 68.3) = 1.75, DF = 5 for MNF and *p* = 0.5041, F(2.82, 71.5) = 0.78, DF = 5 for MDF of the RA muscle; ANOVA *p* = 0.3013, F(3.16, 78.4) = 1.24, DF = 5 for MNF and *p* = 0.1366, F(2.96, 74.0) = 1.90, DF = 5 for MDF of the AM muscle). 

However, the frequencies of the EMG spectrum (reflecting control of motor unit recruitment and muscle fatigue phenomena) showed an increasing trend throughout the session in the RA muscle in particular during the initial session, whereby the Tukey post-test showed significant differences in MNF (*p* = 0.013) and MDF (*p* = 0.029) (Figure 4). 

#### 3.3.3. Effect of the Overall Treatment (Mid-Term Effect)

The effect of the overall treatment on the frequency spectrum of the EMG signal, measured through MNF and MDF, was not significant in the RA muscle (ANOVA *p* = 0.1709, F(2.69, 68.3) = 1.75, DF = 5 for MNF and ANOVA *p* = 0.5041, F(2.82, 71.5) = 0.78, DF = 5 for MDF; comparison in pairs using Tukey post-test was not statistically significant).

On the other hand, the effect of treatment on the frequency spectrum measured through MNF and MDF values did not show an overall significant difference in the AM muscle (ANOVA *p* = 0.3013, F(3.16, 78.4) = 1.24, DF = 5 for MNF and ANOVA *p* = 0.1366, F(2.96, 74.0) = 1.90, DF = 5 for MDF). However, the Tukey paired comparison revealed a significant decrease in MNF at the end of the session between T0 and week 12 (*p* = 0.028; Figure 5). Also, a significant reduction in MDF at the end of the session was evident between T0 and week 6 (*p* = 0.020) and between T0 and week 12 (*p* = 0.018) (see Figure 5).

## 4. Discussion

In the present study, we analyzed the behavior of two muscle groups contributing to the stabilization of the pelvis in a sitting position through control of anteversion and retroversion of the pelvic girdle, stabilization of the backward–forward sway of the trunk, and anchorage of the pelvis while sitting [14,15]. The level and duration of muscle activation throughout execution of a postural task was reflected in the myoelectric characteristics that were recorded and displayed by the use of electromyography. EMG signals, which are complex signals, are deeply influenced by the anatomical and physiological properties of muscles and their alteration, as noticed in patients with CP [34,35,40]. EMG is a common tool often used to study muscle activity with regard to posture and movement, since motion shapes electromyographic recordings [31]. However, very few studies investigating electrical muscle activity and hippotherapy have been published in the scientific literature to date, with even less focusing on patients with CP. The few available studies address the findings from a qualitative point of view, with none carrying out a quantitative analysis of EMG signals from either the time or the frequency domains [18,44,45,46,47]. We focused here on the electrical activity of muscle measured during dynamic postural balance while sitting through surface EMG. We firstly verified the reliability of inter-day EMG measurements in order to validate the methodology, then we analyzed the myoelectrical activation patterns and their evolution with respect to treatment during (i) a short-term interval, i.e., comparing the beginning with the end of each session, and (ii) throughout the whole neurorehabilitation process, i.e., comparing the initial features and their variations after six and twelve weeks of treatment.

A first crucial feature of our results concerned the inter-day reliability of the EMG signal. We noticed that absolute submaximal surface EMG amplitudes obtained during sitting on an unstable support surface (horseback), without external aids, had a substantial inter-day reliability on both measured muscles, as shown by the intraclass correlation coefficient values [48]. Each patient was expected to have some degree of variation in their movements between trials, with CP patients in particular showing significant alterations in muscle control. These data support the assertion that the intersession reliability of our model was sufficient to allow longitudinal analysis of postural muscle behavior in response to hippotherapy.

With this crucial issue verified, a second interesting feature of our results was the symmetry of the muscle action. Previous studies on patients with CP described that impaired muscle tone and, more precisely, spasticity is typically not uniformly spread throughout the trunk and limbs of affected patients. This was stated as a major cause of abnormal body asymmetry between homologous muscle groups, leading to significant postural balance impairment [18,44]. In the present study, some minimal differences were noticed in both the time and frequency domains between the right and the left RA and AM muscles. However, no particular trend emerged, and the differences observed were almost completely attenuated throughout the session, i.e., the left–right asymmetry noticed at the beginning of the treatment (left mRMS was 15% higher than right mRMS, but non-significant) faded at six and twelve weeks. It was conceivable from a physiological point of view that the dynamic smooth solicitation of the postural muscles during hippotherapy harmonized their contractile tempo, in accordance with the postural enhancement during sitting that we usually notice in our patients from a clinical point of view.

Two EMG-related neuromuscular characteristics are usually underlined in patients with CP and can be, to a certain extent, connected to the noticed changes of the present study, helping to explain some of our results. Firstly, excessive EMG intensity of antagonist muscle co-activation during voluntary movement, which was not foreseen here, results from a need to stabilize joints or body segments and seems to be a useful compensatory strategy in patients with CP. Secondly, high-frequency components of the EMG spectrum mainly correspond to high-frequency content of action potentials generated by the fast fiber types (type IIa and IIx fibers) [40]. The higher mean power frequency values observed at the beginning of the hippotherapy treatment probably indicated that during postural balance activities, relatively more fast-twitch muscle fibers were activated. 

When analyzing muscle activation through mRMS, RA remained substantially equivalent (slight decline) during the session regardless of the time of treatment, i.e., at baseline and after six and twelve weeks of the beginning of the therapeutic protocol. The absence of a significant reduction in RMS during the 30-minute session was against the phenomenon of muscle fatigue. This was surprising when considering the relatively poor use of postural muscles in such patients with locomotion and sitting aids (e.g., wheelchairs, molded seats, back braces) and the potential low fatigue resistance of postural muscles in the context of an upper motor neuron lesion [49,50]. Nevertheless, when studying postural muscle activity, any possible change in muscle activation due to fatigue is unlikely to become apparent during recordings of short duration, i.e., 30 minutes. Indeed, the low levels of muscle activity reached in the postural tasks should result in a fatigue time of around several hours [28]. 

Interestingly, the RMS decreased continuously at six weeks (25%) and twelve weeks (30%) with respect to the initial mRMS values. This was coherent with a morphophysiological transformation of the muscle toward a less powerful but more fatigue-resistant muscle following the treatment protocol. Indeed, prolonged endurance training elicits several metabolic and morphological changes, including fast-to-slow fiber-type transformation [51]. This hypothesis was coherent with the behavior of MNF and MDF, whose end-session values decreased gradually in relation to the time of treatment, consolidating the possibility of morphophysiological transformation of postural muscles through hippotherapy. However, it must be noted that after upper motor neuron damage there is marked atrophy and a pronounced shift in the contractile and histochemical profiles from slow-twitch to fast-twitch muscle fibers, with a reduction in oxidative potential and an increase in the fatigability of the affected muscles [52].

Plasticity of skeletal muscle in response to metabolic and functional demands after injury or rehabilitation is driven by metabolic, biochemical, and mechanical properties depending on fiber type. Among the different properties, the myosin heavy chain isoform expression is the most frequently used classification criteria for fiber type [53]. Due to its abundance and contractile significance, qualitative and quantitative changes in myosin and its isoforms exert significant effects on muscle strength and endurance. Human muscles contain three isoforms of myosin heavy chain, called type I, type IIa, and type IIx. A fiber can express a single myosin heavy chain isoform (a pure fiber) or co-express multiple isoforms (a hybrid fiber) [54]. Type I fibers are slow-twitch fibers because of their slow speed of contraction and have a predominantly oxidative metabolism. Type IIx fibers are fast-twitch fibers because of their fast speed of contraction and mainly metabolize glucose via the glycolytic pathway. Type IIa fibers are intermediate fibers with fast contraction speed, but mixed (glycolytic/oxidative) metabolism [55]. Exercise and disuse are the prime determinants of muscle fiber type transition, with the presence of hybrid fibers relating to a high degree of muscle plasticity [56,57]. Alterations in muscle fiber types affect the contractile, metabolic, and biochemical properties of the muscle. The diverse population of muscle fibers in a given muscle allows for various types of tasks, ranging from prolonged, low-intensity contractions (e.g., to maintain posture) to fast and strong maximal contractions. While the fiber type composition of a muscle is genetically determined, muscle plasticity is high and can be stimulated in response to environmental cues, such as targeted functional solicitation [58].

The response to long-lasting physical submaximal exercise, e.g., during a twelve-week hippotherapy cycle, occurs within a very short time and may be linked to changes in concentrations of cellular metabolites which occur after chronic stimulation of skeletal muscle [59]. Mitochondrial biogenesis (whereby mitochondria produce the energy currency of the cell and regulate cellular metabolism) coupled with improved functional parameters of mitochondria play a key role and are a typical response of endurance efforts [60,61]. For instance, it was shown that after only six weeks of training, exercise increases muscle mitochondrial content up to 50%–100% [62]. Among the human muscle fiber types, the mitochondrial content is highest in type I fibers, followed by type IIa and type IIx fibers [63,64]. Mitochondrial changes are intensity-specific, since lower intensity exercise predominantly increases the mitochondrial volume in type I muscle fibers, while recruitment of mitochondria in type II fibers requires much higher intensity exercise [65]. Altogether, endurance exercise positively modulates the overall process as part of muscle adaptation response toward enhanced metabolic and contractile capacity over a very short period of time. The muscle’s oxidative and regenerative potential is thus modulated by increasing the proportion of slow myosin and mitochondria of the muscle fibers [62].

With respect to AM, most cases (11 out of 18 patients) exhibited a bilateral increase in muscle tone, revealed by a spasticity quoted as 2/2+ according to the modified Ashworth scale prior to the start of the protocol. Clinical manifestations corresponded to signs of stiffness as a result of stretch (for example, when adopting the sitting position on the horse). In such patients, the electromyographic activity was not only present during the dynamic phase of stretching, reflecting an archetypal stretch reflex, but persisted during the static phase. In our CP patients, locomotor impairment left the related muscles immobilized in a shortened position leading to muscle contracture, which is a cause of intrinsic hypertonia. Furthermore, muscle immobilization hampers the pivotal mechanism responsible for the development of spasticity, namely, reduction of post-activation depression [66]. Hippotherapy has a direct effect on spasticity [67], providing regular and individualized soft and slow-moving stretching movements along with the correct positioning of limbs. Consequently, a decrease in muscle activation (reflected by mRMS decline) was consistent with decreased muscle tension instead of the appearance of muscle fatigue, where a frequency shift resulted in lower frequencies. This was coherent with the concomitant decrease of MNF and MDF from six weeks after the beginning of the treatment. Nevertheless, the attenuation of muscle spasticity through hippotherapy in our patients was not the only phenomenon that explained the long-term decrease in mRMS and consequently muscle activation. Indeed, one could hypothesize that the significant decrease in mRMS at week 12 was partly the effect of decreased muscle tone and partly fast-to-slow fiber-type transformation consistent with the noticed EMG frequency domain characteristics (MNF and MDF), even if the latter could not be confirmed through our methodology and was beyond the scope of the study.

Taken together, the variations highlighted in the evolution of the myoelectric signal of the selected muscles as a result of the hippotherapy rehabilitation program can be explained in part by two main phenomena: on the one hand, a lesser need for activation of fast-twitch muscle fiber populations in order to stabilize posture and, on the other hand, the stimulation, to some extent, of muscle plasticity mechanisms, leading to a partial oxidative-like reconfiguration of the evaluated muscles.

## 5. Conclusions

Muscle coordination for postural control is basically organized into two functional levels. The first and more basic level deals with the direction specificity of the adjustments. The second deals with fine-tuning of the activated muscles in relation to specificity of the motor action. Interestingly, adjustments of the second level are not triggered by vestibular information, but most likely by a combination of kinesthetic and proprioceptive information from the supporting pelvic region [68], which is one of the main targets in hippotherapy.

In the present work, we focused on two key postural muscles around the pelvic girdle, namely, the rectus abdominis and the adductor magnus. Our aim was to characterize their behavior from a myoelectric point of view by considering the positive clinical effects noticed in postural balance through hippotherapy [10]. However, we did not intend to precisely define muscle fiber type through surface EMG, but rather analyze EMG changes and give some avenues for working hypotheses based on what is known about neuromuscular physiology and previous publications related to patients with cerebral palsy and other neurological conditions.

Our data are fairly consistent with both a more selective recruitment of muscle fiber populations during postural tasks by attenuation of fast-twitch fiber co-contraction and a gradual transformation of muscle contractile properties to optimize the postural profile of key muscles. Moreover, the analysis parameters in the time and frequency domains allowed us to assume that both muscles underwent functional uniformization, reflected by a more symmetrical activation in response to movement of the horse. As a whole, this allowed for more efficient regulation of the output force during postural tasks, which was consistent with our individual clinical observations. The present study provides basic quantitative data supporting the body of scientific evidence determining hippotherapy to be an approach of choice in the neurorehabilitation of patients with CP.

## 6. Study Limitations and Perspectives

This single-group prospective ancillary longitudinal study was conducted in order to evaluate the effects of hippotherapy in the activation and modulation of muscle responses during dynamic postural balance in patients with CP. There is a lack of studies assessing postural control in patients with CP, with many studies consisting of only small sample sizes, reflecting the difficulties that researchers face in recruiting large, homogeneous cohorts from this population, as there is a high variability of clinical features. The “gold standard” in determining the effectiveness of an intervention for clinical validation is a randomized clinical trial protocol, however, increasing numbers of researchers argue that this current standard procedure may be less appropriate for studying complex interventions such as neurorehabilitation. Indeed, this non-drug approach must be carried out according to ethical considerations (difficulty to suspend and/or delay interventions or provide a “placebo procedure” in a control group), intragroup variability (heterogeneous populations with various clinical pictures and levels of disability), interdependent components and contexts, and comprehensive treatments involving organizational complexity and tailored interventions, thereby limiting the methodological validity of the results. The current body of scientific evidence aims to contribute to the creation of rehabilitation programs intended to improve postural and functional skills according to each patient’s needs. However, despite the clear positive clinical effects of hippotherapy with regard to the functional capacities of these patients, which were reinforced by the experimental results of the present study, the absence of a control group potentially makes our conclusions questionable. Finally, no other functional data were collected in this protocol, since the aim of this pilot was to explore the feasibility of the EMG approach in the characterization of postural balance during hippotherapy. The next step will be to correlate such quantitative data changes with changes obtained via validated clinical tests (e.g., the Berg Balance Test, GMFCS, GMFM-66) in randomized clinical trials.

## Figures and Tables

**Figure 1 brainsci-10-00281-f001:**
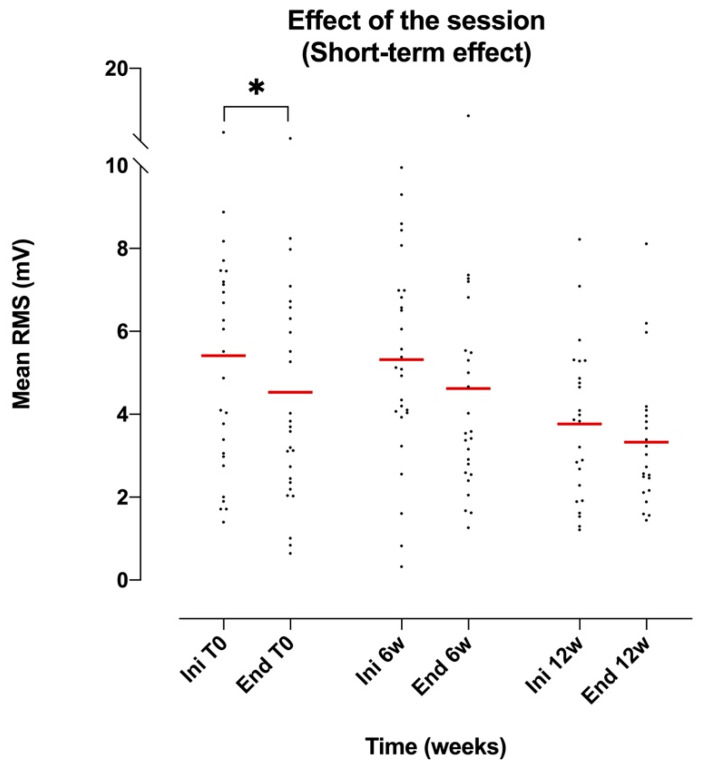
Scatter dot plot of the mean values of the AM mRMS at the beginning and the end of each session (Ini/End) regardless of treatment time. Tukey multiple-comparison test: * (*p* < 0.05).

**Figure 2 brainsci-10-00281-f002:**
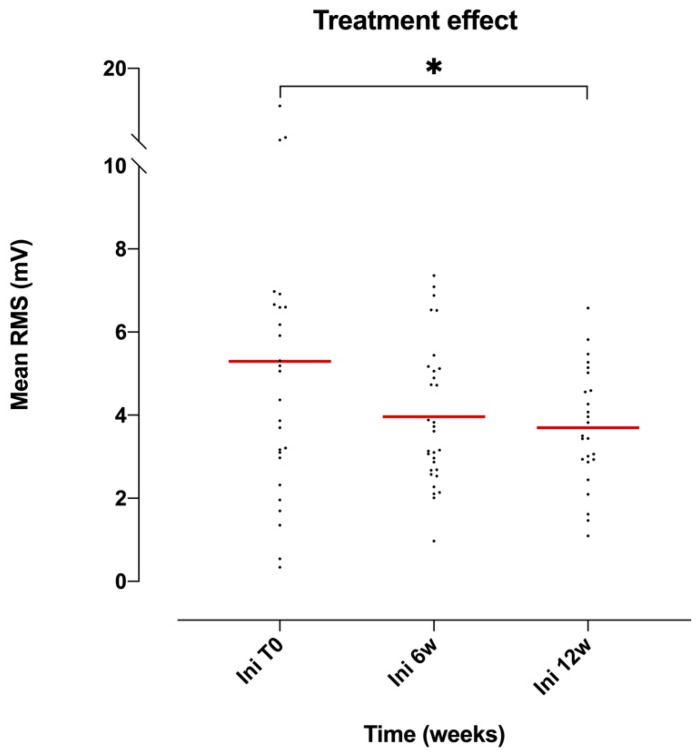
Scatter dot plot of the mean mRMS values of the RA muscle during the first three minutes of the session (Ini) in relation to the treatment time (T0: first session; 6 w: week 6; 12 w: week 12). Tukey multiple-comparison test: * (*p* < 0.05).

**Figure 3 brainsci-10-00281-f003:**
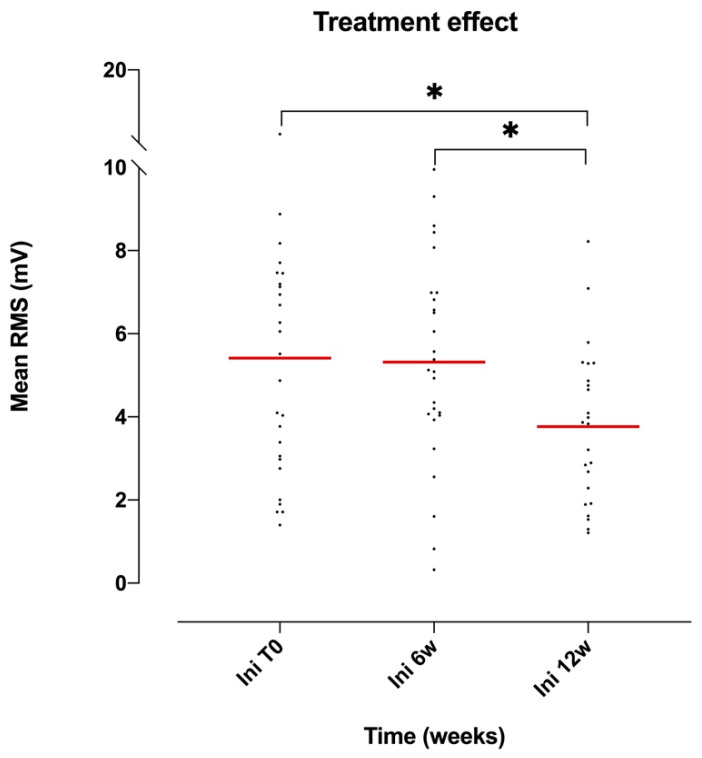
Scatter dot plot around the mean mRMS value of the AM muscle at the first 3 min of the session (Ini) in relation to the treatment time (T0: first session, 6 w: week 6, 12 w: week 12). Tukey multiple-comparison test: * (*p* < 0.05).

**Figure 4 brainsci-10-00281-f004:**
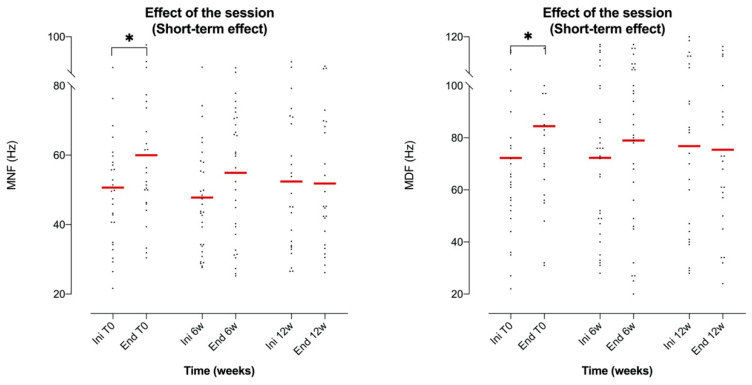
On the left, scatter dot plot of the mean values of RA MNF at the beginning and end of each session (Ini/End) regardless of the treatment time; on the right, scatter dot plot around the mean value of the RA MDF at the beginning and the end of each session (Ini/End) regardless of the treatment time. (T0: first session; 6w: week 6; 12w: week 12). Tukey multiple-comparison test: * (*p* < 0.05).

**Figure 5 brainsci-10-00281-f005:**
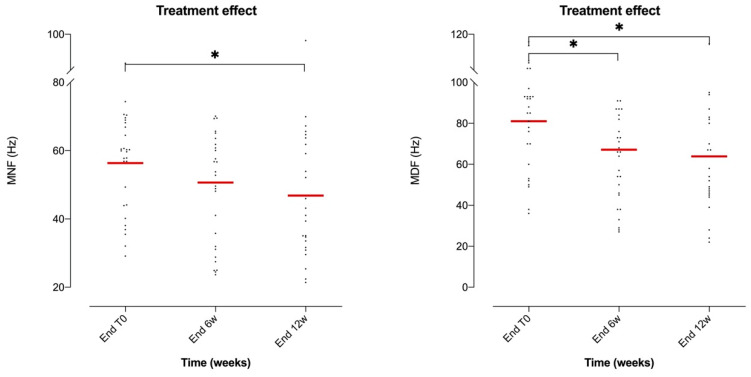
Scatter dot plot of the mean values of MNF (left graph) and MDF (right graph) of the AM muscle with respect to time of treatment (T0: first session; 6w: week 6; 12w: week 12). Data were compared at the end of the session (End). A gradually decrease trend, which was statistically confirmed, emerged at the end of the session in relation to the evolution of the treatment. *: *p* < 0.05 in the Tukey multiple-comparison test.

**Table 1 brainsci-10-00281-t001:** General characteristics of the population.

		F	M	Total
Sex		7	11	18
Age	M	19.4	17.3	18.1
	SD	6.0	5.7	5.7
Motor impairment	Spastic Tetraparesis	5	3	8
	Spastic Paraparesis	-	1	1
	Spastic Diparesis	1	3	4
	Spastic Hemiparesis	-	1	1
	Spastic Monoparesis	-	1	1
	Mixed form	1	2	3
GMFCS	I	-	7	7
	II	-	1	1
	III	4	1	5
	IV	3	2	5
Mild/moderate axial hypotonia *		4	2	6
Spasticity of adductors (2 to 2+) **		5	6	11
Walking patients ***		4	8	12
Standing and walking feasible ****		2	3	5
Wheelchair only *****		3	3	6

* History and physical exam (decreased muscle strength, decreased activity tolerance, delayed motor skills development, rounded shoulder posture, hypermobile joints/increased flexibility); ** modified Ashworth scale; *** walking with or without external aids; **** patients needing external support for transfers and occasionally ting a few steps; ***** patients unable to walk.

**Table 2 brainsci-10-00281-t002:** ANOVA statistics of RA mean root mean square (mRMS) values at the beginning of the session (day 1, week 6 and week 12).

Repeated Measures ANOVA	SS	DF	MS	F (DFn, DFd)	*p*-Value
Treatment (between columns)	39.2	2	19.6	F (1.31, 10.5) = 3.14	*p* = 0.0989
Individual (between rows)	97.2	8	12.2	F (8, 16) = 1.95	*p* = 0.1220
Residual (random)	99.8	16	6.24		
Total	236	26			

**Table 3 brainsci-10-00281-t003:** ANOVA statistics of AM mRMS values at the beginning of the session (day 1, week 6 and week 12).

Repeated Measures ANOVA	SS	DF	MS	F (DFn, DFd)	*p*-Value
Treatment (between columns)	125	2	62.6	F (1.29, 10.3) = 3.36	*p* = 0.0893
Individual (between rows)	271	8	33.9	F (8, 16) = 1.82	*p* = 0.1467
Residual (random)	298	16	18.6		
Total	695	26			

**Table 4 brainsci-10-00281-t004:** Comparison of mRMS values (in mV) for the RA and AM muscles with respect to the side (R: right, L: left), depending of the time of recording (first 3 min and last 3 min of the session) and the time of treatment (initial session, at week 6, and at week 12). Tukey multiple-comparison post-test *p*-values are reported.

	T0 (mV)				Week 6 (mV)				Week 12 (mV)			
	First 3 min		Last 3 min		First 3 min		Last 3 min		First 3 min		Last 3 min	
	R	L	R	L	R	L	R	L	R	L	R	L
RA muscle												
Mean	4.96	5.72	4.31	6.33	3.70	4.24	4.03	3.83	3.86	3.53	3.61	3.48
SD	3.33	4.04	3.57	5.18	1.73	1.67	1.91	1.69	1.41	1.45	1.35	1.45
*p*	0.96		0.55		0.85		0.99		0.97		0.99	
AM muscle												
Mean	5.44	5.38	4.67	4.35	5.92	4.84	5.29	5.89	3.70	3.83	3.40	3.27
SD	3.01	2.86	3.05	2.64	2.19	2.62	4.21	5.03	1.57	2.13	1.67	1.73
*p*	0.99		0.99		0.47		0.94		0.99		0.99	

**Table 5 brainsci-10-00281-t005:** Effect of the session (short-term effect) on the mRMS values (in mV) in the RA and AM muscles at each treatment time (initial session, at week 6, and at week 12). Tukey multiple-comparison post-test *p*-values are reported.

	T0 (mV)			Week 6 (mV)			Week 12 (mV)		
	First 3 min	Last 3 min	*p*	First 3 min	Last 3 min	*p*	First 3 min	Last 3 min	*p*
RA muscle									
Mean	5.30	5.16	0.76	3.96	3.93	0.89	3.70	3.55	0.36
SD	3.61	4.35		1.69	1.77		1.41	1.37	
Confidence interval									
Lower 95%CI	3.87	3.40		3.34	3.28		3.12	2.97	
Upper 95% CI	6.73	6.92		4.58	4.58		4.28	4.13	
AM muscle									
Mean	5.41	4.53	0.01	5.32	4.62	0.24	3.77	3.33	0.15
SD	2.88	2.83		2.46	2.94		1.86	1.66	
Confidence interval									
Lower 95%CI	4.27	3.41		4.35	3.44		2.98	2.59	
Upper 95% CI	6.55	5.65		6.29	5.81		4.55	4.06	

**Table 6 brainsci-10-00281-t006:** Comparison of the mRMS values (in mV) for the RA and AM muscles with respect to the time of treatment (initial value at T0, intermediate value at week 6, and final value at week 12). *p*-values correspond to the ANOVA statistics.

	First 3 min of the Session				Last 3 min of the Session			
	T0	Week 6	Week 12	*p*	T0	Week 6	Week 12	*p*
RA muscle (mV)								
Mean	5.30	3.96	3.70	0.06	5.16	3.93	3.55	0.06
SD	3.61	1.69	1.41		4.35	1.77	1.37	
Confidence interval								
Lower 95%CI	3.87	3.34	3.12		3.40	3.28	2.97	
Upper 95% CI	6.73	4.58	4.28		6.92	4.58	4.13	
AM muscle (mV)								
Mean	5.41	5.32	3.77	0.03	4.53	4.62	3.33	0.03
SD	2.88	2.46	1.86		2.83	2.94	1.66	
Confidence interval								
Lower 95%CI	4.27	4.35	2.98		3.41	3.44	2.59	
Upper 95% CI	6.55	6.29	4.55		5.65	5.81	4.06	

**Table 7 brainsci-10-00281-t007:** Mean frequency (MNF) and median frequency (MDF) of RA muscle (in Hz) with respect to body side at each recording time (first 3 min and last 3 min of the session) and treatment time (T0: first session; 6w: week 6; 12w: week 12). Tukey multiple-comparison post-test *p*-values are reported.

	T0				Week 6				Week 12			
	First 3 min		Last 3 min		First 3 min		Last 3 min		First 3 min		Last 3 min	
	R	L	R	L	R	L	R	L	R	L	R	L
RA MNF (Hz)												
Mean	51.5	45.3	59.6	56.6	48.8	46.7	58.1	51.8	57.7	46.7	56.3	47.4
SD	13.7	17.0	25.1	16.5	15.4	16.8	24.1	19.2	20.8	17.6	21.4	15.3
*p*	0.64		0.98		0.98		0.48		0.07		0.42	
RA MDF (Hz)												
Mean	78.7	58.8	86.5	76.5	78.6	66.1	83.2	74.8	84.2	68.8	82.3	68.5
SD	29.1	23.4	37.3	28.7	29.5	28.5	32.7	31.5	35.4	30.0	35.7	26.2
*p*	0.21		0.69		0.49		0.51		0.12		0.52	

**Table 8 brainsci-10-00281-t008:** MNF and MDF of AM muscle (in Hz) with respect to body side at each recording time (first 3 min and last 3 min of the session) and treatment time (T0: first session; 6w: week 6; 12w: week 12). Tukey multiple-comparison post-test *p*-values are reported.

	T0				Week 6				Week 12			
	First 3 min		Last 3 min		First 3 min		Last 3 min		First 3 min		Last 3 min	
	R	L	R	L	R	L	R	L	R	L	R	L
AM MNF (Hz)												
Mean	53.7	57.9	53.8	59.3	57.9	52.6	51.1	50.3	50.6	45.0	44.4	48.9
SD	19.8	15.8	14.9	13.0	22.4	16.4	15.3	24.1	17.7	20.7	15.6	21.4
*p*	0.92		0.44		0.95		0.99		0.81		0.91	
AM MDF (Hz)												
Mean	74.0	81.1	78.8	83.6	76.3	69.4	66.8	67.4	69.4	62.2	59.5	67.5
SD	28.2	20.4	24.2	21.2	30.0	24.5	19.1	35.3	29.0	35.9	23.8	32.5
*p*	0.95		0.97		0.92		0.99		0.97		0.87	

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
