# Peer review of "Inter-Day Reliability and Changes of Surface Electromyography on Two Postural Muscles Throughout 12 Weeks of Hippotherapy on Patients with Cerebral Palsy: A Pilot Study"

_brainsci, 2020, doi:10.3390/brainsci10050281_

Round 1

Reviewer 1 Report

General comments:

The authors made many improvements based on reviewer feedback.  Methodological and analysis concerns have been addressed. Organization of the results section is improved but could still benefit from further refinement.  There remains some concern regarding interpretation and specifically discussion regarding fiber type changes and other neurophysiological interpretations that are beyond the scope of data was collected or analyzed. 

Specific Comments:

Abstract:

One sentence in the abstract is not appropriate based on the analysis completed. These claims go beyond the scope of the data collected. Please remove the statement on lines 20-22 “Statistical analysis revealed myoelectrical changes of both muscles consistent with a more selective recruitment of muscle fibers and a gradual transformation into predominantly slow-twitch fatigue resistant motor units.”

Information should be provided in the abstract about the reliability of EMG data as shown in both the time domain and frequency domain.  This is the main finding of the study and all that can be claimed based on the current data set.

Introduction:

Reviewer concerns were adequately addressed in this revision

Methods:

Line 122, “non-inclusion” criteria should be replaced by “exclusion” criteria

Line 164-165, the phrase “work on reinforcement of the co-construction of body shema and body image” is not easily understood. Could you replace this with “practiced activities to reinforce internal body image”

The formula on line 276 is not visible in the clean copy of the manuscript. This may not be necessary as long as the reference to Shrout and Fleiss is given.

Results:

The results section is improved and the reliability analysis is a good change.

Please do not use ‘ns’ or ‘*’ for p-values in Tables 2-10, you should include the actual p-value in the table. An * can be used next to any values that reached significance.

Should the data for ANOVA on lines 303 and 307 have Table designation and titles?

 Section 3.3.2 can end after the first paragraph when it is stated that neither muscle showed any significant changes for the short term effect.  Please eliminate lengthy description of non-significant ‘trends’.  This would also eliminate Table 7, Table 8 and figure 4.  

Likewise section 3.3.3 can include just the first paragraph and the rest of that section including Table 9, 10 and figure 5 should be eliminated.  

Discussion:

Three new paragraphs that were added to the discussion (lines 512-516) are not relevant to the data collected or the level of analysis that is possible from the current data set. These paragraphs should be eliminated.

On line 580-581 “partly a fast-to-slow fiber-type transformation consistent with the noticed EMG frequency domain characteristic (MNF and MDF).” Should be deleted. This was not supported by data or analysis from the current data set.

The paragraph on lines 582-587 should be eliminated as it is not supported by the current data set and analysis.

Conclusions:

The paragraph on lines 602-610 should be eliminated from the discussion section as it is not supported by the current data set and analysis.

General recommendations:

While some spelling and grammar errors were corrected some misspellings are still included for example Conexions instead of connections.

Author Response

General comments:

The authors made many improvements based on reviewer feedback. Methodological and analysis concerns have been addressed. Organization of the results section is improved but could still benefit from further refinement. There remains some concern regarding interpretation and specifically discussion regarding fiber type changes and other neurophysiological interpretations that are beyond the scope of data was collected or analyzed.

Thank you for your comments and advice. Indeed, some elements of the discussion are not factual since the characterization of fiber type changes cannot be confirmed nor the more selective recruitment of the studied muscles. However, in our opinion these two reasons can contribute to put into perspective the overall EMG results. Our aim here is to bring into the light some ideas that could be developed further. We believe they are important in the context of the present work. However, since the interpretations are, as the reviewer mentions it, beyond the scope of collected data, we tried to rephrase them since we do not want to over-interpret our results.

Specific Comments:

Abstract:

One sentence in the abstract is not appropriate based on the analysis completed. These claims go beyond the scope of the data collected. Please remove the statement on lines 20-22 “Statistical analysis revealed myoelectrical changes of both muscles consistent with a more selective recruitment of muscle fibers and a gradual transformation into predominantly slow-twitch fatigue resistant motor units.”

Information should be provided in the abstract about the reliability of EMG data as shown in both the time domain and frequency domain. This is the main finding of the study and all that can be claimed based on the current data set.

The abstract was rewritten accordingly, emphasizing the demonstrated reliability of EMG data that is the main finding of the study. We tried to give also a potential explanation to the EMG changes over time, not as a direct interpretation of data but as a hypothesis to deal with.

Introduction:

Reviewer concerns were adequately addressed in this revision

Methods:

Line 122, “non-inclusion” criteria should be replaced by “exclusion” criteria

The expression was corrected accordingly.

Line 164-165, the phrase “work on reinforcement of the co-construction of body shema and body image” is not easily understood. Could you replace this with “practiced activities to reinforce internal body image”

“work on reinforcement of the co-construction of body schema and body image” has been replaced by “practiced activities to reinforce internal body image” accordingly.

The formula on line 276 is not visible in the clean copy of the manuscript. This may not be necessary as long as the reference to Shrout and Fleiss is given.

Since the Word Equation was not correctly included, we have included the formula as an image.

Results:

The results section is improved and the reliability analysis is a good change.

Please do not use ‘ns’ or ‘*’ for p-values in Tables 2-10, you should include the actual p-value in the table. An * can be used next to any values that reached significance.

We have replaced ‘ns’ and ‘*’ by their actual p-values. We have also corrected the Tables’ legends accordingly. In tables 5 and 6, we deleted the information concerning the confidential intervals that was not necessary.

Should the data for ANOVA on lines 303 and 307 have Table designation and titles?

We preferred to consider the data for ANOVA as a text since it was lighter for the reader. For this reason, no Table designation and titles are given.

Section 3.3.2 can end after the first paragraph when it is stated that neither muscle showed any significant changes for the short term effect. Please eliminate lengthy description of non-significant ‘trends’. This would also eliminate Table 7, Table 8 and figure 4.

We have deleted the description of non-significant trends as well as Tables 7 and 8. Since, there are some significant differences, we have kept Figure 4.

Likewise section 3.3.3 can include just the first paragraph and the rest of that section including Table 9, 10 and figure 5 should be eliminated.

We have deleted Tables 9 and 10. We have kept the paragraphs where statistical significances have been noticed. We have also kept Figure 5 for the same reasons.

Discussion:

Three new paragraphs that were added to the discussion (lines 512-516) are not relevant to the data collected or the level of analysis that is possible from the current data set. These paragraphs should be eliminated.

The paragraph has been deleted accordingly.

On line 580-581 “partly a fast-to-slow fiber-type transformation consistent with the noticed EMG frequency domain characteristic (MNF and MDF).” Should be deleted. This was not supported by data or analysis from the current data set.

We have rephrased the paragraph in order to put into perspective the results with respect to one possible physiological explanation, avoiding an emphatic position leading to an over-interpretation of the results. We think is important to provide some avenues to be explored further.

The paragraph on lines 582-587 should be eliminated as it is not supported by the current data set and analysis.

Again, it is not our intention to over-interpret our data. Thus, we have rephrased the paragraph in order to summarize our point of view.

Conclusions:

The paragraph on lines 602-610 should be eliminated from the discussion section as it is not supported by the current data set and analysis.

We agree that the conclusions are not directly supported by the current data set and analysis. However, as evoked hereinbefore, or aim here was to provide some avenues for future studies. This is explained in the second paragraph of the conclusion “However, we did not intend to precisely define muscle fiber type through surface EMG but to analyze EMG changes and give some avenues for work hypotheses based on what is known about neuromuscular physiology and what has been published related to patients with cerebral palsy and other neurological conditions.”

In order to take into account the remarks of the reviewer, we tried to rephrase the last paragraph in order to propose a language allowing the reader to remain open-minded (Cf. Reviewer 2) and trying to minimize the over-interpretation of our results.

General recommendations:

While some spelling and grammar errors were corrected some misspellings are still included for example Conexions instead of connections.

We have double-checked for spelling and grammar errors and have corrected some unnoticed misspellings (format: English USA).

Reviewer 2 Report

Quality of the manuscript has been substantially improved since the previous submission. Authors attended to all comments made by the reviewers, and provided adequate additional information.  I think data provided in the manuscript can be used by others for sample size calculations and determination of MDC. Findings might have been over-interpreted, but the language used allows the reader to remain open-minded. There are minor editing points to which I refer below:

  1. Results, section 3.1, first and second tables: please attend to the commas that should replace full stops in reporting F statistic DF.
  2. Line 476: intra-session perhaps should be replaced by inter-session.
  3. Line 602: Data is singular. Please replace are with is.

Author Response

Quality of the manuscript has been substantially improved since the previous submission. Authors attended to all comments made by the reviewers, and provided adequate additional information. I think data provided in the manuscript can be used by others for sample size calculations and determination of MDC. Findings might have been over-interpreted, but the language used allows the reader to remain open-minded. There are minor editing points to which I refer below:

  1. Results, section 3.1, first and second tables: please attend to the commas that should replace full stops in reporting F statistic DF.

Full stops were duly replaced by comas in section 3.1 ANOVA tables.

  1. Line 476: intra-session perhaps should be replaced by inter-session.

Intra-session was replaced by inter-session accordingly.

  1. Line 602: Data is singular. Please replace are with is.

Grammar has been corrected accordingly.

Round 2

Reviewer 1 Report

I am satisfied with the changes made by the authors.

This manuscript is a resubmission of an earlier submission. The following is a list of the peer review reports and author responses from that submission.

Round 1

Reviewer 1 Report

General comments:

This is an interesting and relevant topic. There are a number of methodological and analysis concerns that could raise questions about the interpretation. The results section would benefit from improved organization and reduction in duplication of data.

Specific Comments:

Abstract:

Several terms are used in the abstract (i.e. myoelectrical shape, predominantly fatigue resistant motor units) that are not adequately justified or explained in the body of the manuscript

Introduction:

Page 1, lines 33-34: should include information that the clinical picture may also be influenced by life experience and activity level.

Throughout the manuscript, there are a number of informal statements such as “Fine-tuning of postural muscles is “hard” (line 46), please be more explicit. 

Page 2, lines56-58.  Please add a reference for information about the type of movements produced during hippotherapy mimicking that of the patient’s body during walking. Is this true for gait patterns associated with CP or is it mimicking the patterns of a typically developing person’s body during walking?

Please provide more justification and explanation for the choice of muscles for EMG.  Why not monitor trunk extensors as well as flexors?  Adductor magnus is a deep muscle and there are several superficial muscles that would be closer to the surface electrode and that could overlay the signal coming from the adductor magnus.  How was cross talk from other muscles eliminated?

Methods:

Please provide more information regarding methods.   

Several of the inclusion criteria could be challenging for patients with CP with functional levels of GMFCS III and IV.  How was “normal” hip joint function assessed? How were impairments in sensorimotor or cognitive function evaluated and confirmed?

Detailed explanation of the exercise procedure was clear and clinically relevant.  However the phrase “work on integration different segments in the body diagram” was confusing. Please restate this in a way that is more transparent.  Please explain why stirrups were not used?  Wouldn’t it help the patient stabilize better?

How were electrodes placed with respect to boney landmarks and soft tissue?   How was the patient placed during electrode placement?  There is a large age range and large range of functional ability. How was electrode placement altered for different ages to assure correct anatomical placement?  Was EMG collected while the patient was sitting on the horse?  If so, how was activity of the horse standardized across time and across patients?  How were changes in skin resistance accounted for with the EMG signal if the electrodes were worn for the entire session, especially with warmth and rubbing on the adductor magnus due to position on the horse? How were motion artifacts dealt with considering the pressure on the AM surface electrode with horse’s body?  Was there difficulty keeping the electrode in place during the riding session?  

Please provide more justification for the decision to use the outcomes chosen for the EMG data.  The choice of outcome measures for EMG used references that are more than 30 years old and for studies that were conducted on healthy adults.  Have these methods been shown to be valid and reliable for recording data from muscles for patients with CP?  Are there more recent studies using these parameters?  Why or why not?   It is not valid to compare for left-right asymmetry because the results depend on exact placement of the surface EMG?  Likewise, amplitude measurements across sessions might be related to slightly altered alignment of the sensors.  Please justify the decision to use the specific outcomes chosen and verify the reliability and validity of these outcomes across time and repeated placement of electrodes. 

Were the EMG data normalized? Was a maximal voluntary contraction attempted?  If so, how accurate would this be for patients with CP?

The study population information (lines 204-211) belongs in the methods section with description of the participants.  Please move it to methods.

Table 1:  What does +/+++ or +/++ mean? You report 18 patients, 10 wheel chair users and 12 with standing and walking ability.  This implies that at least 6 patients could not stand and walk. Is this correct? How were the patients placed on the horse if they could not stand or walk?

Results:

Please specify units for data in Tables.

Figure legends should be numbered and clearly labeled.  The Table legend should be on the final row of the table not above it.

Could changes across time during each session be explained by increased warmth and decrease in skin resistance due to duration of wearing the electrodes for 30 minutes?  

Please select either a table or a figure, do not present the data redundantly in both a table and a figure.  There are too many figures, please select two or three of the most pertinent results and create plots for them. Other results can be described or included in a table. 

Were any functional data collected?  If so, how did they relate to the EMG findings?

Discussion:

The discussion should include information about other studies that used these outcome measures and demonstrated that they were reliable and valid across repeated measures in humans.  The references used seem weakly related to the statements being made. Please provide more specific references that demonstrate that this technique and interpretation are justified.

Page 17, lines 423-441.  The discussion of fiber type is speculative at best.  The ability to define fiber type from surface EMG is controversial.  The speculation that EMG results represent the “possibility of morphophysiological transformation of postural muscles”. Is this physiologically likely over the short time periods in the study?  Is there evidence that this could be verified by surface EMG?

Conclusions:

While the information in the first paragraph of the conclusion is theoretically interesting, it is not specifically related to the procedure and hypotheses of this study.  Dorsal muscles were NOT tested, so there can be no comparison of dorsal vs. ventral muscle activation patterns.  Please remove this paragraph.

In the second paragraph the authors mention the desire to characterize the reshape of EMG that might account for positive clinical effects.  Were clinical effects on postural balance documented for the patients in this study?  If so, it might be possible to compare change in clinical measures with change in physiological measures. Please report these clinical measures if they were collected.

General recommendations:

Spelling and grammar check, e.g. connexions should be connections, symetric should be symmetric,

People first language “CP patients” should be “patients with CP”

Reviewer 2 Report

Thank you very much for giving me the opportunity to read this manuscript. The study examined the effect of a 12-week hippotherapy intervention on the pattern of muscle activity of two postural muscles in individuals with CP. Authors employed mean Root Mean Square (mRMS), median (MDF) and mean (MNF) frequency contents of the Rectus Abdominis (RA) and Adductor Magnus (AM) muscles to quantify changes in the magnitude of muscle activity and possible transformation in the muscles fibre composition based on the frequency spectrums of the EMG throughout the course of intervention. This was an exploratory study and hence no research hypothesis was suggested, and subsequently tested.

There were major design issues with the study (stated below), based on which I recommend rejection of the manuscript in the current form. However, difficulties in the recruitment of participants with CP and dearth of data collected from this population, might have worked in favour of publication of this study if it was written in a different form.

Reliability of the measurements were not assessed by the authors. Use of surface electromyography over several sessions during such dynamic activity, requires providing a measure for the reliability of the data, and an estimation of the Minimal Detectable Change (MDC) for the measured variables. This could have been followed by a wash-out period with no hippotherapy session, after which, the main intervention study could have been completed. This way, authors could have partially addressed lack of a control group. As authors have already identified, lack of a control group further weakens inferences made based on the findings.

As stated above, although I do not think this manuscript is publishable in the current form, authors may decide to take a different approach and try publishing the paper in a different journal. For example, data collected over the course of the intervention, can be used as pilot data. Authors may estimate values of the MDC for the measured variables based on the data and write their manuscript around it.    

There were other issues around this manuscript:

Participants completed 12 weeks of training. There were 2 sessions per week and 30 minutes of training per session: i.e. overall 12 hours of training over 12 weeks. Fibre transformation has been suggested by the authors as a possible mechanism underlying observed changes in mRMS and frequency spectrums at the termination of the study. Available literature in human and animal models, suggest chronic training (e.g. 12 hours of electrical stimulation over several weeks) as the requisite stimulus for muscle fibre type transformation. Whether training time was of long enough duration to support fibre transformation was not discussed by the authors.

Statistical reports were not complete. Reporting F statistic and DoF in reporting outcomes of the ANOVA can help the reader on the accuracy of the analysis conducted.

Although this is a writing style preference, I found the first two paragraphs of the Discussion section irrelevant, and for the sake of brevity, this section could have been started from line 403. The same criticism is also applicable to the Conclusion section. This section could have been started from line 478. Please note movement is misspelled in line 481.